# Improvement of the Chemical Reactivity of Michael Acceptor of Ethacrynic Acid Correlates with Antiproliferative Activities

**DOI:** 10.3390/molecules28020910

**Published:** 2023-01-16

**Authors:** Abdelmoula El Abbouchi, Nabil El Brahmi, Marie-Aude Hiebel, Hamza Ghammaz, Elmostafa El Fahime, Jérôme Bignon, Gérald Guillaumet, Franck Suzenet, Saïd El Kazzouli

**Affiliations:** 1Euromed Research Center, Euromed Faculty of Pharmacy, Euromed University of Fes (UEMF), Meknes Road, Fez 30000, Morocco; 2Institut de Chimie Organique et Analytique, Université d’Orléans, UMR CNRS 7311, BP 6759, CEDEX 2, 45067 Orléans, France; 3Centre National de la Recherche Scientifique et Technique (CNRST), Angle Avenues des FAR et Allal El Fassi, Hay Ryad, Rabat 10102, Morocco; 4Institut de Chimie des Substances Naturelles, CNRS, Université Paris-Saclay, 91190 Gif-sur-Yvette, France

**Keywords:** ethacrynic acid, anticancer, synthesis, antiproliferative, α, β-unsaturated ketone, Michael acceptor, MTS

## Abstract

The present study aims to report the design, synthesis, and biological activity of new ethacrynic acid (**EA**) analogs (**6–10**) obtained by the double modulation of the carboxylic acid moiety and the aromatic ring with the aim to increase the chemical reactivity of Michael acceptor of **EA**. All obtained compounds were characterized by ^1^H and ^13^C NMR, IR, and high-resolution mass spectrometry. The antiproliferative activity was evaluated in vitro using MMT test, in a first step, against HL60 cell line and in a second step, on a panel of human cancer cell lines such as HCT116, A549, MCF7, PC3, U87-MG, and SKOV3, and normal cell line MRC5 in comparison with positive control doxorubicin. Among all the tested compounds, the product **8** containing a propargyl and a hydroxyl groups, allowing an intramolecular hydrogen bond with the keto group of **EA**, exhibited a pronounced and selective activity in a nanomolar range against HL60, A549, PC3, and MCF7 with IC_50_ values of 15, 41.2, 68.7, and 61.5 nM, respectively. Compound **8** also showed a good selectivity index (SI) against HL60 and moderate SI against the other three human cancer cells (A549, PC3, and MCF7). The study of the structure-activity relationship showed that both modifications of the carboxylic group and the introduction of an intramolecular hydrogen bond are highly required to improve the antiproliferative activities. The molecular modeling studies of compound **8** revealed that it favorably binds to the glutathione S-transferase active site, which may explain its interesting anticancer activity. These new compounds have potential to be developed as novel therapeutic agents against various cancer types.

## 1. Introduction

Currently, cancer is the leading cause of death in high and upper-middle income countries and the second most common cause of death worldwide [1]. In 2020, the World Health Organization (WHO) estimated that almost 10 million people died from cancer and 19.3 million new cases were diagnosed in the same year [2]. Hence, developing new small molecules with high specificity in order to combat this disease is a big challenge for researchers.

The α, β-unsaturated carbonyl groups are attractive in medicinal chemistry due to their chemical properties, biological activities, and therapeutic potential, especially in oncology [3].

Several natural and synthetic compounds, such as chalcones [4], ibrutinib [5,6], canertinib [7,8], afatinib [9,10], neratinib [11,12], hypothemycin [13], and their analogues, are known for their antitumor activities, thanks to their α, β-unsaturated carbonyl which acts as a Michael acceptor. This system can easily react with nucleophiles, such as thiol of cysteine under physiological conditions [14]. Therefore, the increase of the reactivity of the Michael acceptor group has currently attracted more attention [15]. This electrophile site is present in ethacrynic acid (**EA**), which is thought to be responsible for its excellent inhibitory properties of the glutathione S-transferase P1-1 (GSTP1-1) [16]. The GSTP1-1 is actively involved in the detoxification of cells and even in the elimination of many cytotoxic agents, which represents a major obstacle to anticancer chemotherapy. Additionally, it has anti-apoptotic activities based on protein–protein interactions with the *N*-terminal c-Jun kinase (JNK) by directly blocking the function of JNK [17,18]. In fact, most anticancer agents induce cell death by activating the mitogen-activated protein–kinase (MAPK) pathways, particularly those involving JNK and p38-MAPK [19]. There is sufficient evidence to suggest that GSTP1-1 inhibitors may be useful therapeutic agents for the treatment of cancer and other diseases associated with aberrant cell proliferation. **EA** analogs, alone or as adjuvants, are known to increase the activity of chemotherapeutic agents [20,21,22,23]. Based on this observation, we have considered improving the reactivity of the Michael acceptor group of **EA** to increase the inhibitory activity of GSTP1-1, on the one hand, and on the other hand to increase the antiproliferative activity via the apoptosis process through the activation of the caspase protease as we have demonstrated in our previous work [24,25].

Regarding all these above-mentioned factors, we have considered synthesizing new analogs of **EA**, by replacing the chlorine at the *ortho* position of the α, β-unsaturated carbonyl group with a hydroxyl group (to favorize the formation of an intramolecular hydrogen bond) and by removing the chlorine in the meta position. At the same time, the carboxylic acid group should be transformed on either amide (G1) or propoargylic group (G2) which have shown good activities in our previously published works [26,27,28,29]. Consequently, the reactivity of the Michael acceptor of **EA** could be enhanced by the establishment of an intramolecular hydrogen bond between the carbonyl and the hydroxyl group (Figure 1). All the prepared compounds were tested first toward HLC60 cell line, then **7** and **8** compounds have been selected for antiproliferative studies against a panel of tumor cell lines such as human adenocarcinomic epithelial cell line (A549), human breast cancer cell line (MCF7), human prostate cancer cell line (PC3), human glioblastoma (U87-MG), ovarian carcinoma cells (SKOV3), and human colon cancer cell line (HCT116).

## 2. Results and Discussions

### 2.1. Chemistry

The synthetic route of the targeted **EA** analogue **6** is outlined in Figure 1. In the first step, the compound **2** was prepared in 76% yield via a Friedel–Crafts reaction from **1** and butyryl chloride in the presence of titanium tetrachloride (TiCl_4_) as a Lewis acid in dichloromethane (DCM) [30]. Subsequently, the second step involves an aldol condensation reaction in the presence of formaldehyde and potassium carbonate in a mixture of ethanol/water (1/1) for 24 h to give compound **3** in 44% yield. Compound **4** as the main precursor for the synthesis of **EA** derivatives, was obtained under demethylation conditions using aluminum chloride (AlCl_3_) in anhydrous DCM in 78% yield. Next, the dihydroxyl compound **4** is involved in a Williamson-type selective nucleophilic substitution reaction in the presence of potassium iodide, tert-butyl-2-bromoacetate, and potassium tert-butoxide (*t*-BuOK) yielding compound **5** in 63%. It is very important to note that when ethyl ester and methyl ester were used substrates instead of *tert*-butyl ester **5**, their hydrolysis under basic reaction conditions led to intramolecular cyclisation between the hydroxyl group and the Michael receptor. In the final step, the ester group of **5** is successfully converted into its corresponding carboxylic acid group using trifluoroacetic acid (TFA) in DCM at room temperature for 12 h to afford the target molecule **6** with a good yield (89%).

After developing the synthesis of compound **6** as a new **EA** derivative, we were interested thereafter in carrying out two main structural modifications starting from the intermediate **4**. Firstly, we have introduced the butyl acetamide function on the 4-hydroxy group to prepare an amide derivative of compound **6**. In this case, the desired compound 7 was obtained via a nucleophilic substitution reaction (*SN2*) by reacting the compound **4** with the 2-bromo-*N*-butylacetamide, which was prepared and used without further purification according to the literature [31]. Secondly, following the same reactivity a propargylic group was introduced on the 4-hyrdroxy group by using propargyl bromide in the presence of *t*-BuOK and KI in THF at 60 °C for 24 h which afforded compound **8** in 41% yield.

In order to compare the effect of the substitution of the chlorine by the hydroxyl group on the antiproliferative activities, the analogs **9** and **10** were synthetized according to Figure 2. Compound **9** was isolated in a 72% yield via an amidation reaction of **EA** with butan-1-amine in the presence of EDC and HOBt as activating agents through a mechanism similar to the previously reported in the literature [32]. Firstly, **EA** was deprotonated by EDC, then a nucleophilic attack of the protonated EDC gives a good leaving group which facilitates the nucleophilic attack of HOBt in the next step, giving **EA** derivative containing HOBt as the leaving group. Finally, the attack of the amine gave the desired product **9** and urea and HOBt as by-products. Compound **10** was prepared as previously reported [23] (Figure 2). For more detail and data of the synthesized compounds, see the supplementary materials.

### 2.2. Biological Study

Based on a multiple target assay, all the prepared compounds (**6–10**) were first evaluated against HL60 leukemia cells to assess their in vitro antiproliferative activities. As mentioned in the Table 1, the obtained results were expressed as half-maximal inhibitory concentration (IC_50_) values with the average of at least three independent experiments and are compared to **EA**.

From the screening results in Table 1, compound **6** exhibited weak antiproliferative activity with an IC_50_ value > 10 µM, showing that the modification of the aromatic ring without modifying the carboxylic group did not improve the antiproliferative activity of the **EA**. Effectively, once the carboxylic acid function of the compound **6** is modified by an amide function (compound **7**), an interesting antiproliferative activity was observed with an IC_50_ = 1.7 μM. Interestingly, its analog (compound **9**), which bears two chlorines on the aromatic ring, remains inactive. These results confirm the positive impact of the hydroxyl group on the biological activity and bode well with the development of new **EA** analogs containing a hydroxyl group in *ortho* position of the Michael acceptor. Interestingly, compound **8** exhibited a very potent antiproliferative activity with an IC_50_ = 0.015 µM, despite the absence of the amide function [22]. Furthermore, in agreement with the positive effect of the substitution of the chlorine atoms by a hydroxyl group, compound **8** shows markedly improved antiproliferative activity in comparison with compound **10** (IC_50_ value of 0.58 µM) (Figure 2) [25].

Based on these results, we have evaluated the antiproliferative activity of the two selected compounds **7** and **8** against a panel of cancer and non-cancer cell lines representative of diverse tissues/organ tumors: human adenocarcinomic epithelial cell line (A549), human breast cancer cell line (MCF7), human prostate cancer cell line (PC3), human glioblastoma (U87-MG), ovarian carcinoma cells (SKOV3), and human colon cancer cell line (HCT116) using cell survival assays. Moreover, in order to evaluate the safety index, their effect on the proliferation of human lung fibroblasts (MCR5) as a normal cell line was also studied. The cytotoxicities of compounds **7** and **8** (expressed as IC_50_ values) obtained after 72 h of exposure are summarized in Table 2.

The data shown in Table 2 indicates that for A549 and MCF7 cells, compound **8** was found to be more active than both compound **7** and doxorubicin, used as a reference, with IC_50_ values of 41.2 and 61.5 nM, respectively. We have also noticed in the case of PC3 and U87-MG cells that compound **8** was more active than compound **7** but less active than doxorubicin with IC_50_ values of 68.7, 369, and 2.09 nM, respectively. Conversely, we have observed in the case of HCT116 that compound **7** is slightly more active than compound **8,** but both were less active than doxorubicin with IC_50_ values of 920, 1260, and 90 nM, respectively.

Based on these very encouraging results, the selectivity index (SI) of compounds **7** and **8** was calculated for each cancer cell lines (SI, defined as the ratio of the IC_50s_ of non-cancer cells/IC_50s_ of cancer cells) (Table 3) [33,34]. The results have corroborated the well-known poor selectivity of doxorubicin and have emphasized the selectivity of our compounds, especially compound **8** which exhibits a good SI against HL60 (SI = 23.33) and moderate ones on the other three cancer cells (A549, MCF7, and PC3). Unfortunately, compound **7** is less selective compared to compound **8** on all cancer cells. Next, the classical physicochemical parameter such as the membrane–water partition coefficient (LogP) was evaluated, using Marvin application ChemAxon for the two compounds **7** and **8** in order to correlate it with their antiproliferative activities. The lipophilicity, which combines both biological and physicochemical properties, is a crucial property for drug absorption, distribution, potency, and elimination. The compounds **7** and **8** showed almost the same LogP with the values of 3.44 and 2.67, respectively, suitable for the future development of drug-like compounds [35].

Then, we decided to achieve docking molecular studies using the best anticancer agent **8** and the glutathione S-transferase as target enzyme. With compound **8,** we obtained the nine different conformations from docking analysis, from which the eighth conformer showed the best results with a total of 10 bonds, the first three being the closest to the Glutathione S-transferase active site (1.84 Å, 3.05 Å, and 3.57 Å respectively). The one at 3.05 Å from the active site showed the presence of a hydrogen bond with tyrosine at the position 108 of the pocket, as well as a weak hydrogen bond with glycine at the position 205. The rest are hydrophobic alkyl bonds, with a pi-pi stacked hydrophobic bond with phenylalanine at position 8 of the active site. The docking analysis shows that compound **8** favorably binds to the glutathione S-transferase and further explains the interesting anticancer results, especially, against the HL60 cancer cell line (Figure 3).

## 3. Materials and Methods

### 3.1. General Procedures

All reagent-grade chemicals and solvents commercially available were used without further purification. The reactions were monitored by thin-layer chromatography (TLC) using aluminum (Kiesel gel 60F254, Merck, Darmstadt, Germany) and visualized using ultraviolet light (λ = 254 nm or 365 nm). Flash column chromatography was performed on silica gel 60 (230–400 mesh, 0.040–0.063 mm). Melting points (mp (°C)) were carried out by open capillary tubes and are uncorrected using Thermo scientific digital melting point IA9200. IR absorption spectra were obtained on a Thermo Scientific Nicolet iS10, and the values are reported in inverse centimeters. ^1^H NMR and ^13^C NMR spectra were recorded on a Bruker AC300 (300 MHz for ^1^H and 75 MHz for ^13^C) or on a Bruker 400 (400 MHz for ^1^H and 100.6 MHz for ^13^C) spectrometer at room temperature, and the samples dissolved in an appropriate deuterated solvent. The peak patterns are indicated as follows: s, singlet; d, doublet; t, triplet; m, multiplet; q, quartette. The coupling constants, *J*, are reported in Hertz (Hz). HRMS spectra were acquired in positive mode with an ESI source on a Q-TOF mass spectrometer with an accuracy tolerance of 2 ppm.

### 3.2. Synthesis and Characterization

The compounds **2** and **10** were prepared according to the methodology described in the literature [25,30,36].

1-(2,4-Dimethoxyphenyl)-2-methylenebutan-1-one (**3**). To a mixed solution of water and ethanol (17 mL/10 mL), compound 2 (1.1 g, 5 mmol) in ethanol (34 mL), 30% solution of formaldehyde (0.5 mL, 5 mmol), and potassium carbonate (0.67 g, 5 mmol) were added. The mixture was stirred for 24 h at 60 °C. To the reaction mixture, a hydrochloric acid solution (a mixture of 2 mL of concentrated hydrochloric acid and 100 mL of water) was added; then, the medium was decanted, extracted with (3 × 10 mL) ethyl acetate, dried over magnesium sulfate, filtered, and concentrated to dryness. The crude residue was purified by chromatography on a silica gel column using hexane/EtOAc, 7: 3 (*v/v*) as eluent, to obtain the expected product 3 under form of a white solid with a yield of 44%. Mp: 109–113 °C. ^1^H NMR (400 MHz, CDCl_3_), δ 7.30 (d, *^3^J_HH_* = 8.3 Hz, 1H), 6.56-6.45 (m, 2H), 5.72 (s, 1H), 5.56 (s, 1H), 3.86 (s, 3H), 3.79 (s, 3H), 2.47 (q, *^3^J_HH_* = 7.4 Hz, 2H), 1.13 (t, *^3^J_HH_* = 7.4 Hz, 3H). ^13^C NMR (101 MHz, CDCl_3_), δ 197.7, 162.7, 159.3, 151.7, 131.3, 124.2, 122.1, 104.1, 98.8, 55.6, 55.4, 24.1, 12.5. IR: ν (cm^−1^): = 1661 (C=O Ketone). HRMS (+ESI) *m/z*: [M+H]+ calculated for C13H17O3: 221.1171, found, 221.1172.

1-(2,4-Dihydroxyphenyl)-2-methylenebutan-1-one (**4**)**.** A solution of anisole **3** (5 g, 22.5 mmol) in DCM (120 mL) was cooled to −78 °C, then an amount of AlCl_3_ (1 M in DCM, 149 mL, 149 mmol) was added dropwise over 10 min. The reaction mixture was stirred at 40 °C for 24 h. After total consumption of the limiting reagent, the reaction was stopped by the addition of methanol (60 mL). The reaction medium was concentrated to dryness then taken up in ethyl acetate and washed several times with an aqueous solution of HCl (1 M). The organic phases were dried over magnesium sulfate, filtered, and concentrated to dryness. The crude residue was purified by chromatography on a silica gel column using hexane/EtOAc, 7: 3 (*v/v*) as eluent, to obtain the expected product **4** as liquid form with a yield of 78%. ^1^H NMR (400 MHz, CDCl_3_), δ 12.64 (s, 1H), 7.68 (d, *^3^J_HH_* = 8.8 Hz, 1H), 6.92 (br, 1H), 6.45 (d, *^3^J_HH_* = 2.4 Hz, 1H), 6.41 (dd, *^3^J_HH_* = 2.4, 8.8 Hz, 1H), 5.58 (s, 1H), 5.32 (s, 1H), 2.47 (t, *^3^J_HH_* = 7.4 Hz, 2H), 1.12 (t, *^3^J_HH_* = 7.4 Hz, 3H). ^13^C NMR (101 MHz, CDCl_3_), δ 203.1, 165.9, 163.6, 148.4, 135.6, 119.0_,_ 113.1, 107.8_,_ 103.6_,_ 26.5, 12.1. IR: ν (cm^−1^): = 3333 (OH), 1669 (C=O Ketone). HRMS (+ESI) *m/z*: [M+H]^+^ calculated for C_11_H_13_O_3_: 193.0856, found: 193.0859.

*Tert*-butyl 2-(3-hydroxy-4-(2-methylenebutanoyl)phenoxy)acetate (**5**)**.** To a solution of phenol **4** (0.92 g, 3 mmol) in THF, potassium *tert*-butoxide (0.33 g, 5 mmol), a catalytic amount of KI (0.05 g, 0.3 mmol), and *tert*-butyl bromoacetate (0.58 g, 3 mmol) were added, and the whole mixture was stirred at 60 °C for 24 h. After cooling to room temperature, a solution of concentrated HCl was added to pH = 1, and then the mixture was extracted three times with ethyl acetate. The organic phases were washed with brine, dried over magnesium sulphate, filtered, and then concentrated to dryness. The crude residue was purified by chromatography on a silica gel column using DCM/EtOAc, 4: 1 (*v/v*) as eluent, to obtain the expected product **5** as a white solid with a yield of 63%. Mp. 125–126 °C. ^1^H NMR (400 MHz, CDCl_3_),), δ 12.59 (s, 1H), 7.70 (d, *^3^J_HH_* = 8.9 Hz, 1H), 6.47 (dd, *^3^J_HH_* = 8.9, *^4^J_HH_* = 2.5 Hz, 1H), 6.41 (d, *^3^J_HH_* = 2.5 Hz, 1H), 5.59 (s, 1H), 5.32 (s, 1H), 4.57 (s, 2H), 2.47 (q, *^3^J_HH_* = 7.4 Hz, 2H), 1.52 (s, 9H), 1.13 (t, *^3^J_HH_* = 7.4 Hz, 3H). ^13^C NMR (101 MHz, CDCl_3_), δ 202.6, 167.0, 166.0, 164.4, 148.5, 134.7, 119.0, 113.6, 107.3, 101.8_,_ 82.9, 65.4, 28.0 (3C), 26.4, 12.1. IR: ν (cm^−1^): = 3312 (OH), 1651 (C=O Ketone), 1586 (C=O Ester). HRMS (+ESI) *m/z*: [M+H]^+^ calculated for C_17_H_23_O_5_: 307.1538, found: 307.1500.

2-(3-Hydroxy-4-(2-methylenebutanoyl)phenoxy)acetic acid (**6**). To a solution of ester **5** (0.08 g, 0.261 mmol) in DCM (10 mL), trifluoroacetic acid (6 mL) was added, and the mixture was stirred at room temperature for 12 h. The resulting mixture was concentrated, and the residue was dissolved in DCM (1 mL), then the desired product **6** was precipitated by the technique of co-evaporation using diethyl ether. The expected product **6** was obtained as a white solid with a yield of 89%. Mp. 62–164 °C. ^1^H NMR (400 MHz, CDCl_3_), δ 12.58 (br s, 1H), 10.72 (br s, 1H), 7.72 (d, *^3^J_HH_* = 8.9 Hz, 1H), 6.49 (dd, *^3^J_HH_* = 8.9, *^4^J_HH_* = 2.5 Hz, 1H), 6.45 (d, *^3^J_HH_* = 2.5 Hz, 1H), 5.60 (s, 1H), 5.33 (s, 1H), 4.74 (s, 2H), 2.47 (q, *^3^J_HH_* = 7.4 Hz, 2H), 1.12 (t, *^3^J_HH_* = 7.4 Hz, 3H). ^13^C NMR (101 MHz, CDCl_3_), δ 202.7, 173.1, 165.9, 163.8, 148.4, 134.9, 119.2, 113.9, 107.1, 101.9_,_ 64.5, 26.3, 12.1. IR: ν (cm^−1^): = 3402 (OH), 1652 (C=O Ketone), 1622 (C=O Acid). HRMS (+ESI) *m/z*: [M+H]^+^ calculated for C_13_H_15_O_5_: 251.0910, found: 251.0914.

*N*-Butyl-2-(3-hydroxy-4-(2-methylenebutanoyl)phenoxy)acetamide (**7**). To a solution of phenol **4** (0.57, 3 mmol) in tetrahydrofuran, potassium *tert*-butoxide (0.33 g, 5 mmol), a catalytic amount of KI (0.05 g, 0.3 mmol), and 2-bromo-*N*-butylacetamide (0.58, 3 mmol) were added. The reaction mixture was heated at 60 °C for 24 h. After cooling to room temperature, a solution of concentrated HCl was added to pH = 1, and then the mixture was extracted three times with ethyl acetate. The organic phases were combined, washed with brine, dried over magnesium sulphate, filtered, and concentrated to dryness. The crude residue was purified by chromatography on a silica gel column using DCM/EtOAc, 4: 2 (*v/v*) as eluent, to obtain the expected product **7** as a white solid with a yield of 56%. Mp. 133–134 °C. ^1^H NMR (400 MHz, CDCl_3_), δ 12.52 (s, 1H), 7.73 (d, *^3^J_HH_* = 8.9 Hz, 1H), 6.57–6.43 (m, 3H), 5.61 (s, 1H), 5.33 (s, 1H), 4.54 (s, 2H), 3.37 (q, *^3^J_HH_* = 7.4 Hz, 2H), 2.47 (q, *^3^J_HH_* = 7.4 Hz, 2H), 1.59–1.53 (m, 2H), 1.42–1.32 (m, 2H), 1.13 (t, *^3^J_HH_* = 7.4 Hz, 3H), 0.95 (t, *^3^J_HH_* = 7.4 Hz, 3H). ^13^C NMR (101 MHz, CDCl_3_), δ 202.7, 167.0, 165.9, 163.3, 148.5, 135.0, 119.3, 114.1, 106.4, 102.7_,_ 67.2, 38.9, 31.6, 26.3, 20.0, 13.7, 12.1. IR: ν (cm^−1^): = 3350 (OH), 1663 (C=O Ketone), 1602 (C=O Amide). HRMS (+ESI) *m/z*: [M+H]^+^ calculated for C_17_H_24_NO_4_: 306.1700, found: 306.1699.

1-(2-Hydroxy-4-(prop-2-yn-1-yloxy)phenyl)-2-methylenebutan-1-one (**8**). To a solution of phenol **4** (0.69 g, 3 mmol) in tetrahydrofuran, potassium *tert*-butoxide (0.33 g, 5 mmol), a catalytic amount of KI (0.05 g, 0.3 mmol) and propargyl bromide (0.35 g, 3 mmol) were added. The reaction mixture was heated at 60 °C for 24 h. After cooling to room temperature, a solution of concentrated HCl was added to pH = 1 and then the mixture was extracted three times with ethyl acetate. The organic phases were combined, washed with brine, dried over magnesium sulphate, filtered, and concentrated to dryness. The crude residue was purified by chromatography on a silica gel column using Hexane/DCM, 1: 1 (*v/v*) as eluent, to obtain the expected product **8** as liquid form with a yield of 41%. ^1^H NMR (300 MHz, CDCl_3_), δ 12.6 (s, 1H), 7.71 (d, *^3^J_HH_* = 8.8 Hz, 1H), 6.57 (d, *^3^J_HH_* = 2.4 Hz, 1H), 6.49 (dd, *^3^J_HH_* = 2.4, 8.8 Hz, 1H), 5.99 (s, 1H), 5.32 (s, 1H), 4.75 (d, *^4^J_HH_* = 2.4 Hz, 2H), 2.59 (t, *^4^J_HH_* = 2.4 Hz, 1H), 2.47 (q, *^3^J_HH_* = 7.4 Hz, 2H), 1.12 (t, *^3^J_HH_* = 7.4 Hz, 3H). ^13^C NMR (75 MHz, CDCl_3_), δ 202.7, 165.9, 164.0, 148.6, 134.7, 119.0_,_ 113.6_,_ 107.4_,_ 102.4, 76.4, 56.0, 56.0, 26.4, 12.1. IR: ν (cm^−1^): = 3289 (OH), 2122 (C≡C Alkyne), 1678 (C=O Ketone). HRMS (+ESI) *m/z*: [M+H]^+^ calculated for C_14_H_15_O_3_: 231.0975, found: 231.0977.

*N*-Butyl-2-(2,3-dichloro-4-(2-methylenebutanoyl)phenoxy)acetamide (**9**). To a solution of EDC (0.56 g, 3.63 mmol), HOB*t* (0.55 g, 3.63 mmol), DMAP (0.02 g, 0.16 mmol), and ethacrynic acid (0.10 g, 3.30 mmol) in DMF anhydrous (5 mL), butan-1-amine (0.24 g, 3.30 mmol) was added at 0 °C, and the whole mixture was stirred overnight at room temperature. The mixture was extracted with ethyl acetate, and the combined organic phases were washed with brine, dried over magnesium sulfate, and then concentrated under pressure. The crude residue was purified by chromatography on a silica gel column using DCM/EtOAc, 4: 1 (*v/v*) as eluent, to obtain the expected product **9** as a white solid with a yield of 72%. Mp. 98–100 °C. ^1^H NMR (400 MHz, CDCl_3_), δ 7.19 (d, *^3^J_HH_* = 8.5 Hz, 1H), 6.87 (d, *^3^J_HH_* = 8.5 Hz, 1H), 6.78 (br s, 1H), 5.96 (s, 1H), 5.59 (s, 1H), 4.57 (s, 2H), 3.39 (q, *^3^J_HH_* = 7.4 Hz, 2H), 2.47 (q, *^3^J_HH_* = 7.4 Hz, 2H), 1.63–1.53 (m, 2H), 1.47–1.32 (m, 2H), 1.15 (t, *^3^J_HH_* = 7.4 Hz, 3H), 0.95 (t, *^3^J_HH_* = 7.4 Hz, 3H). ^13^C NMR (101 MHz, CDCl_3_), δ 195.5, 166.5, 154.5, 150.2, 134.1, 131.4, 128.7, 127.2, 122.9, 110.9_,_ 68.2, 38.9, 31.5, 23.4, 20.0, 13.7, 12.4. IR: ν (cm^−1^): 1660 (C=O Ketone), 1599 (C=O Amide). HRMS (+ESI) *m/z*: [M+H]^+^ calculated for C_17_H_22_Cl_2_NO_3_: 358.0973, found: 358.0971.

### 3.3. Biological Evaluation

All cancer cell lines (HCT116, A549, MCF7, PC3, U87-MG, SKOV3, and MRC5) were obtained from the European Collection of Cell Culture (ECACC, Salisbury, UK) or from the American Type Culture Collection (Rockville, MD, USA) and were cultured according to the supplier’s instructions. Human HCT-116 colorectal carcinoma, MCF7 breast adenocarcinoma, PC3 prostate adenocarcinoma, A549 lung carcinoma, and SK-OV-3 ovary carcinoma were grown in Gibco McCoy’s 5A supplemented with 10% fetal calf serum (FCS) and 1% glutamine. HL60 myelogenous leukemia cells were grown in RPMI 1640 supplemented with 10% fetal calf serum (FCS) and 1% glutamine. U87-MG glioblastoma and human MRC-5 cells were grown in Gibco medium DMEM supplemented with 10% fetal calf serum (FCS) and 1% glutamine. Cells were maintained at 37 °C in a humidified atmosphere containing 5% CO_2_. Cell growth inhibition was determined by an MTS assay according to the manufacturer’s instructions (Promega, Madison, WI, USA). Briefly, the cells were seeded in 96-well plates (2.5 × 10^3^ cells/well) containing 100 μL of growth medium. After 24 h of culture, the cells were treated with the tested compounds at 10 different final concentrations. After incubation for 72 h, 20 µL of Cell Titer 96^®^AQ_ueous_. One Solution Reagent was added for 2 h before recording absorbance at 490nm with a spectrophotometric plate reader. The concentration–response curves were generated with Graph Prism software and the 50% inhibition concentration values (IC_50_) were calculated from polynomial curves (four or five-parameter logistic equations).

### 3.4. Molecular Modeling

Molecular docking was performed using the Autodock Vina [37,38]. AutoDockTools was required to prepare the PDBQT file for Glutathione S-transferase and to set the size and the center of the grid box. Kollman charges, Gasteiger charges, AD4 type atom, and polar hydrogen atoms were included in the Glutathione S-transferase structure. The grid size was set at 30 × 30 × 30 (x, y, and z) points, and the grid center was designated at x, y, and z dimensions of 19.951167, −0.126697, and 20.556417, respectively. Post-docking analyses were visualized using Discovery Studio Biovia 2021, which showed the sizes and locations of binding sites, hydrogen-bond interactions, hydrophobic interactions, and bonding distances at 5 Å from the position of the docked ligand; compound being docked to the active site of Glutathione S-transferase. Subsequently, binding poses of each ligand were observed and their interactions with the protein were characterized, and the best conformations of each ligand were selected.

## 4. Conclusions

In summary, new **EA** analogues have been synthesized by modifying two parts of the **EA**, namely the carboxylic acid part and the aromatic ring part. All the synthesized compounds were evaluated and screened for their in vitro antiproliferative activities first against HL60. Based on their activities, two selected compounds, 7 and 8, were tested on a panel of human cancer cell lines and on a normal cell line. Interestingly, compound 8 shows very potent anti-proliferative activity against the four human cancer cells HL60, A549, PC3, and MCF7 with IC_50_ values of 15 nM, 41.2 nM, 68.7, and 61.5 nM, respectively. Compound 8 also shows a good SI against HL60 and moderate SI on the other three cancer cells (A549, PC3, and MCF7). This study reports the crucial and positive impact of an intramolecular hydrogen bond to increase the chemical reactivity of the Michael acceptor of the **EA** analogs, which correlates with the improvement of the antiproliferative activities.

## Data Availability

All data supporting this research are availible Appendix A.

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
