# Peer review of "Improvement of the Chemical Reactivity of Michael Acceptor of Ethacrynic Acid Correlates with Antiproliferative Activities"

_molecules, 2023, doi:10.3390/molecules28020910_

Round 1

Reviewer 1 Report

The authors described the chemical reactivity of Michael acceptor of ethacrynic acid correlates with antiproliferative activities. From an organic point of view, the scope is exceedingly limited. In addition, the results of anticancer activity are not highlighted in comparison to the standard drug (doxorubicin). I consider that docking molecular studies might be included. In summary, I consider that manuscript does not meet all requirements to be published in Molecules (Q1). Additional suggestions and comments are included: (1) See Table 2. The antiproliferative activity of compounds 6-10 should be included. ¿What is the reason to choose only compounds 7 and 8? However, these results are not very good in comparison to the standard drug (doxorubicin). (2) See Scheme 2. I consider that the scope of compounds 7, 8, and 9 is exceedingly limited. It should be increased to obtain more examples, followed by their anticancer evaluation.    (3) See Table 3. I consider that the selectivity index is not relevant because the results of anticancer activity are not good. (4) The anticancer results should be complemented by docking molecular studies. It is mandatory.

Author Response

The authors described the chemical reactivity of Michael acceptor of ethacrynic acid correlates with antiproliferative activities. From an organic point of view, the scope is exceedingly limited. In addition, the results of anticancer activity are not highlighted in comparison to the standard drug (doxorubicin). I consider that docking molecular studies might be included. In summary, I consider that manuscript does not meet all requirements to be published in Molecules (Q1). Additional suggestions and comments are included:

  • See Table 2. The antiproliferative activity of compounds 6-10 should be included. ¿What is the reason to choose only compounds 7 and 8? However, these results are not very good in comparison to the standard drug (doxorubicin).

Response: Thank you dear referee 1 for your questions. In this study we have tested two families of compounds. The first one contains compounds 7 and 9 and the second family contains 8 and 10 compounds. The antiproliferative activity of all compounds was tested first on HLC60 cells (Table 1) and then we have chosen that the most actives compounds were 7 and 8. Then we decided to test them on other cancer cell lines (Table 2). When we compare the obtained results to the doxorubicin we can observe that the compound 8 is selective to certain types of cancers. 

  • See Scheme 2. I consider that the scope of compounds 7, 8, and 9 is exceedingly limited. It should be increased to obtain more examples, followed by their anticancer evaluation.

Response: We completely agree with you. With this synthesis method we can prepare many examples as we have published (https://doi.org/10.1016/j.bmcl.2020.127426 ; https://doi.org/10.1016/j.bioorg.2021.105293) . However, our aims in this article is to study the relation between the Michael acceptor activation (when we replace chlorine by hydroxyl), and the biological activity. The proof of our concept was validated by comparing 7 and 8 with 9 and 10, respectively.

  • See Table 3. I consider that the selectivity index is not relevant because the results of anticancer activity are not good.

Response: Thank you reviewer 1 for this comment, I don’t agree with you because our best compound 8 has an IS of 23.33 for MRC5/HL60 while doxorubicin has an IS of 3.98.

  • The anticancer results should be complemented by docking molecular studies. It is mandatory.

Response: Thank you again for this relevant comment, I agree with you and I added an interesting doking study between compound 8 and the glutathione S-transferase (see revised version).

Reviewer 2 Report

The work on the “ Improvement of the Chemical Reactivity of Michael Acceptor of Ethacrynic Acid Correlates with Antiproliferative Activities”  is a valuable perfect, and suitable contribution to be published in Molecules Journal after justifying some points.

 1-     Abstract

·       the in vitro should be written in italics line 16

·       Line 16 the sentence seems unclear with first, …

·       Line 18 you can write in comparison with positive control doxorubicin.

·       Write IC50 with as IC50 subscript

·       Line 21 you can write nM in the last IC50 value no need to repeat it many times.

·       The abstract is too short, it should be improved by adding the values of IC50 and the main findings.

·       You can add a sentence regarding the chemical characterization methods of these compounds “ NMR, HRMS, IR “?? To abstract.

·       Add the main used biological methods

·       Add a conclusion and future planned works

2-     Introduction

·       The introduction is well written but may you can improve it by adding the problem statement (cancer disease) in the first paragraph

·       You can add some statistical data regarding cancer to the introduction from recent publications.

·        add a paragraph in the introduction regarding the biological planes that will be conducted and the cell line names

3-     Results and discussion

·       This section was well written but some points should be discussed more

·       Can the authors discuss why they used HOBt/EDC reagents for amidation and mention another method?? A recent publication has used this method to make better yield you can cite it as https://doi.org/10.1186/s13065-021-00793-8, to improve the discussion, moreover, you can mention and discuss the mechanism of action of EDC which was discussed recently https://doi.org/10.1186/s13065-022-00839-5, all of this recent publication can improve the discussion section of this work

·       Table 1 how could you calculate the IC50 values just by using one conc. (1 uM) ?? correct the cation of this table

·       the footnote of Table one contain “na: not actif” correct it as not active

·       Regarding the selectivity index can you improve this section and collected data with reference (https://doi.org/10.1515/hc-2020-0134) regarding this ratio calculation between normal cell lines and cancer cell lines ??  

·       Lines 160-179 regarding LogP, Lipinski rule of Five, and drug score, can you improve these results with all parameters and factors of drug-likeness and RO5, by using similar work to this published article “Synthesis, chemo-informatics, and anticancer evaluation of fluorophenyl-isoxazole derivatives” you can make a table and include all the parameters of your compounds to present these finding well, you can use the Molsoft (http://www.molsoft.com/) and Molinspiration (http://www.molinspiration.com/), to predict the bioactivity score and molecular properties of newly designed compounds.

4-     Materials and methods

·       Lines 189-205 take your effort to improve these paragraphs to reduce the percentage of similarity.

·       I could not see the chemical data of NMR nor HRMS in the Main MS or the supp. file ??

·       Lines 312-329 this paragraph was copied as it is from one reference, I know that is a procedure but you can re-write it again to reduce the

·       The General information of the chemical synthesis was copied as it is without any changes, it should be rewritten for the percentage of similarity.

5-     Control again all references as the journal style  

 Best wishes 

Author Response

The work on the “ Improvement of the Chemical Reactivity of Michael Acceptor of Ethacrynic Acid Correlates with Antiproliferative Activities”  is a valuable perfect, and suitable contribution to be published in Molecules Journal after justifying some points.

 1-     Abstract

  • the in vitro should be written in italicsline 16

Response: Thank you dear referee 2 for your comments. The requested modification is added to the revised version in yellow color

  • Line 16 the sentence seems unclear with first, …

Response: The requested modification is added to the revised version in yellow color

  • Line 18 you can write in comparison with positive control doxorubicin.

Response: The requested modification is added to the revised version in yellow color

  • Write IC50 with as IC50 subscript

Response: The requested modification is added to the revised version in yellow color

  • Line 21 you can write nM in the last IC50 value no need to repeat it many times.

Response: The requested modification is added to the revised version in yellow color

  • The abstract is too short, it should be improved by adding the values of IC50 and the main findings.

Response: The requested modification is added to the revised version in yellow color  

  • You can add a sentence regarding the chemical characterization methods of these compounds “NMR, HRMS, IR “?? To abstract.

Response: The requested modification is added to the revised version in yellow color

  • Add the main used biological methods
  • Add a conclusion and future planned works

Response: Thank you referee 2 for this all comments. We have added your comments on the abstract with yellow color 

2-     Introduction

  • The introduction is well written but may you can improve it by adding the problem statement (cancer disease) in the first paragraph.
  • You can add some statistical data regarding cancer to the introduction from recent publications.
  • add a paragraph in the introduction regarding the biological planes that will be conducted and the cell line names.

Response: Thanks, we have added your comments in the introduction in a yellow color

3-     Results and discussion

  • This section was well written but some points should be discussed more
  • Can the authors discuss why they used HOBt/EDC reagents for amidation and mention another method?? A recent publication has used this method to make better yield you can cite it as https://doi.org/10.1186/s13065-021-00793-8, to improve the discussion, moreover, you can mention and discuss the mechanism of action of EDC which was discussed recently https://doi.org/10.1186/s13065-022-00839-5, all of this recent publication can improve the discussion section of this work ok

Response: Thank you referee 2 for this interesting comment. The proposed references were added.

Concerning the activation of carboxylic acid, we have used first HOBt/DCC as activator agents but we had purification problems to remove the urea which is a by-which comes from DCC. For that reason, we have used HOBt/EDC as an alternative.

  • Table 1 how could you calculate the IC50 values just by using one conc. (1 uM) ?? correct the cation of this table.

Response: Thank you referee 2 for this interesting comment, it was a mistake on the title of this table. The IC50 were calculated using various concentrations (the title of the table was corrected in the revised version)

  • the footnote of Table one contain “na: not actif” correct it as not active

Response: Thanks, your request modification is in yellow color

  • Regarding the selectivity index can you improve this section and collected data with reference (https://doi.org/10.1515/hc-2020-0134) regarding this ratio calculation between normal cell lines and cancer cell lines ????????
  • Lines 160-179 regarding LogP, Lipinski rule of Five, and drug score, can you improve these results with all parameters and factors of drug-likeness and RO5, by using similar work to this published article “Synthesis, chemo-informatics, and anticancer evaluation of fluorophenyl-isoxazole derivatives” you can make a table and include all the parameters of your compounds to present these finding well, you can use the Molsoft (http://www.molsoft.com/) and Molinspiration (http://www.molinspiration.com/), to predict the bioactivity score and molecular properties of newly designed compounds. ????????

Response: Thank you referee 2 for this comment, in order to improve our paper and as you mention predict the biological activity, we added a new paragraph on the docking molecular studies.

4-     Materials and methods

  • Lines 189-205 take your effort to improve these paragraphs to reduce the percentage of similarity.
  • I could not see the chemical data of NMR or HRMS in the Main MS or the supp. file

All the characterization data were mentioned in Materials and Methods part and all the spectrums are in the supporting information part

  • Lines 312-329 this paragraph was copied as it is from one reference, I know that is a procedure but you can re-write it again to reduce the

Response: Thank you reviewer 2 for this comment all the modifications you request were done in the revised version

  • The General information of the chemical synthesis was copied as it is without any changes, it should be rewritten for the percentage of similarity.

Response: Thank you reviewer 2 for this comment all the modifications you request were done in the revised version

5-     Control again all references as the journal style 

Response: All the references were checked according the journal style

Reviewer 3 Report

Comments:

In this manuscript, the authors have successfully synthesized a series of new ethacrynic acid analogues to increase the chemical reactivity.

In vitro experiments on HL60 cell lines and on several human cancer cell lines showed that derivative which contains a propargyl and a hydroxyl groups showed a noticeable and selective activity in a nanomolar concentration range compare with doxorubicin used as reference.  Authors explain this activity considering the formation of intramolecular H-bond. I find the new data interesting, supported by appropriate characterization of the new compounds, and their antiproliferative activity well-presented. Taking all the above into account, I propose to accept the manuscript for publication in its present form.

Author Response

In this manuscript, the authors have successfully synthesized a series of new ethacrynic acid analogues to increase the chemical reactivity.

In vitro experiments on HL60 cell lines and on several human cancer cell lines showed that derivative which contains a propargyl and a hydroxyl groups showed a noticeable and selective activity in a nanomolar concentration range compare with doxorubicin used as reference.  Authors explain this activity considering the formation of intramolecular H-bond. I find the new data interesting, supported by appropriate characterization of the new compounds, and their antiproliferative activity well-presented. Taking all the above into account, I propose to accept the manuscript for publication in its present form.

Response: Thank you reviewer 3 for your time and effort reviewing our paper.

Round 2

Reviewer 1 Report

The authors described the chemical reactivity of Michael acceptor of ethacrynic acid correlates with antiproliferative activities. From an organic point of view, the scope is exceedingly limited. According to my previous revision, I consider that manuscript does not meet all requirements to be published in Molecules (Q1) in terms of low scientific rigor and relevance of the results presented here. However, the final decision will depend on the Editor.  

Author Response

The authors described the chemical reactivity of Michael acceptor of ethacrynic acid correlates with antiproliferative activities. From an organic point of view, the scope is exceedingly limited. According to my previous revision, I consider that manuscript does not meet all requirements to be published in Molecules (Q1) in terms of low scientific rigor and relevance of the results presented here. However, the final decision will depend on the Editor.

Response to referee 1. We would like to thank you for your time and effort and we respect your decision.

Reviewer 2 Report

The work on the “Improvement of the Chemical Reactivity of Michael Acceptor of Ethacrynic Acid Correlates with Antiproliferative Activities” was well improved and suitable to be published in Molecules Journal after justifying some minor points.

·        in the Abstract “NMR, HRMS, IR characterization method” this sentence should not be added after the results it should be before.

·        The introduction was well improved accordingly

·        The Results and discussion section was well improved but the  a recent publication https://doi.org/10.1186/s13065-021-00793-8, regarding these reagents it is better than 2007 old article, and to improve the discussion, moreover, you can mention and discuss the mechanism of action of EDC which was discussed recently https://doi.org/10.1186/s13065-022-00839-5, all of this recent publication can improve the discussion section of this work ok.

·        Regarding the selectivity index can you improve this section and collected data with reference https://doi.org/10.1515/hc-2020-0134, regarding this ratio calculation between normal cell lines and cancer cell lines, you have to cite reference regarding these calculation

·        Regarding figure 3 edit the figure with better resolution because the table in this figure is unclear as well as edit the legend of this figure and add compound 8.  

·        all methods should be cited with references because this is not a new method.

Best wishes  

Author Response

The work on the “Improvement of the Chemical Reactivity of Michael Acceptor of Ethacrynic Acid Correlates with Antiproliferative Activities” was well improved and suitable to be published in Molecules Journal after justifying some minor points.

  • in the Abstract “NMR, HRMS, IR characterization method” this sentence should not be added after the results it should be before.

Response: Thank you for this comment, the modification was made on the revised version.

  • The introduction was well improved accordingly
  • The Results and discussion section was well improved but the  a recent publication https://doi.org/10.1186/s13065-021-00793-8, regarding these reagents it is better than 2007 old article, and to improve the discussion, moreover, you can mention and discuss the mechanism of action of EDC which was discussed recently https://doi.org/10.1186/s13065-022-00839-5, all of this recent publication can improve the discussion section of this work ok.

Response: Thank you for this comment, the modification was made on the revised version.

  • Regarding the selectivity index can you improve this section and collected data with reference https://doi.org/10.1515/hc-2020-0134, regarding this ratio calculation between normal cell lines and cancer cell lines, you have to cite reference regarding these calculation.

Response: Thank you for this comment, the modification was made on the revised version.

  • Regarding figure 3 edit the figure with better resolution because the table in this figure is unclear as well as edit the legend of this figure and add compound 8. 

Response: Thank you for this comment, the modification was made on the revised version.

  • all methods should be cited with references because this is not a new method.

Response: Thank you for this comment, the modification was made on the revised